# Effects of Dietary Inclusion of Tannin-Rich Sericea Lespedeza Hay on Relationships among Linear Body Measurements, Body Condition Score, Body Mass Indexes, and Performance of Growing Alpine Doelings and Katahdin Ewe Lambs

**DOI:** 10.3390/ani12223183

**Published:** 2022-11-17

**Authors:** Wei Wang, Amlan Kumar Patra, Ryszard Puchala, Luana Ribeiro, Terry Allen Gipson, Arthur Louis Goetsch

**Affiliations:** 1College of Animal Science and Veterinary Medicine, Shenyang Agricultural University, Shenyang 110866, China; 2American Institute for Goat Research, School of Agriculture and Applied Sciences, Langston University, Langston, OK 73050, USA; 3Department of Animal Nutrition, West Bengal University of Animal and Fishery Sciences, Kolkata 700037, India

**Keywords:** body condition score, body mass index, diet quality, forage type, linear body measures, species

## Abstract

**Simple Summary:**

Body condition score (BCS), linear body measures (height, length, and circumference), and body mass indexes (BMI) are useful tools for livestock feeding management and breeding practices. This study assessed relationships among these measures as well as with performance variables of body weight (BW), dry matter intake (DMI), and BW gain (ADG) in Katahdin (KAT) ewe lambs vs. Alpine (ALP) doelings fed diets varying in quality. Correlations between BMI and BW, DMI, and ADG were greater than those involving BCS, and ones for BCS of KAT were greater than for ALP. Correlations involving BMI based on combinations of two linear measures (wither height, hook or pin bone length, and heart girth) with BW and ADG were greater than for BMI based one or three linear measures. Thus, BMI based on two linear measures compared with the BCS may be a better livestock management tool because of being relatively less subjective than BCS.

**Abstract:**

The objective of this study was to assess the effects of the dietary level of a condensed tannin-rich forage on linear measures, body condition score (BCS), body mass indexes (BMI), and performance and relationships among these variables in growing dairy goats and hair sheep raised for meat. An experiment with a 2 × 3 factorial treatment arrangement was conducted, with two species and three diets. Diets were 25% concentrate and 75% forage, which were alfalfa hay, condensed tannin-containing Sericea lespedeza hay, and a 1:1 mixture of both hay sources. Twenty-four Alpine (ALP) doelings and 24 Katahdin (KAT) ewe lambs were used in the 173-day study, consisting of four measurement periods. Variables included BCS, linear measures, BMI, and performance variables such as average daily gain (ADG) and dry matter intake (DMI, g/day). Linear measures were length from the shoulder point to pin (Pin) and hook (Hook) bones, height at the withers (Wither), circumference from heart girth (Heart), and width at the hook bones (Rump). Different BMI were based on the Wither, Hook, Pin, Heart, and various combinations. Heart, Rump, and all BMI were affected by species, whereas linear measures and BMI based on Wither and the combination of height and length measures were influenced by diet. There were positive (*p* < 0.05) correlation coefficients (r) between BCS and body weight (BW) and linear measures as well as BMI for both species, and correlations were greater for KAT than for ALP. Body condition score was correlated with BW, ADG, and DMI for KAT (*p* < 0.05). For ALP, the correlation between BCS and BW (*p* < 0.015) was much lower than for KAT (0.49 vs. 0.91), and there were only tendencies for relationships between BCS and ADG and DMI (*p* < 0.10). Body weight and ADG were positively correlated (*p* < 0.05) with all BMI for both species, and most correlations were greater for KAT than for ALP. Correlations involving BMI based on combinations of two linear measures with BW and ADG were in most instances greater than for BMI calculated from one or three linear measures. For ALP, DMI was related to BMI based on Wither and Pin (r = 0.43), Heart and Hook (r = 0.44), and Heart and Pin (r = 0.61), whereas for KAT, correlations were similar (0.72–0.75) for each of the four BMI based on two linear measures (i.e., Wither and Hook, Wither and Pin, Heart and Hook, and Heart and Pin). Therefore, it appears that each of these four BMI, preferably based on Heart and length, could be appropriate for better livestock management over BCS and in predicting animal performance.

## 1. Introduction

Different morphological traits, body condition score (BCS), and body mass indexes (BMI) are common tools for livestock management and breeding practices [1,2,3,4,5,6]. Such measures have been traditionally used to characterize local genetic resources and for breed classification [1,7] as well as for predicting body weight (BW) and market value [6,8,9,10,11,12,13]. Linear body measures include heart girth, body length, wither height, and rump height and width. The total size of animals is a function of body length, height, and circumference [14,15]. Heart girth alone or combined with body length and wither height measurements in Malabari female adult goats predicted BW with good accuracy [16]. Genetic correlations of BW with length, heart girth, and wither height were high in Jamunapari goats, indicating the scope of genetic selection for higher BW using these linear body measurements [17]. Heart girth and length were found to be appropriate predictors of BW and selection indicators to improve genetic merit in BW of sheep and goats [1,10,18].

Body condition score is used to evaluate nutritional status and obesity and determine appropriate feeding management practices during critical physiological periods [3,5]. Body condition score is more highly predictive of body fat content [19,20] compared with BW [20,21]. However, BCS is somewhat subjective and prone to inherent inaccuracy due to differences in fat deposition sites [22,23,24]. Use of BW relative to linear measurements to estimate BMI may yield less subjective assessments of the nutritional plane and health status [4,23,25]. For example, feeding palmitic acid and rapeseed oil resulted in differences in BW and BMI but not BCS [22]. Moreover, BMI appeared to be superior to BCS for predicting meat and milk production of Wassachian sheep [4]. A BMI was found to be more highly associated with effects of short-term energy withdrawal on pulsatile luteinizing hormone secretion in goats relative to BW [26]. In yearling Alpine doelings fed forage-based diets, there were stronger relationships between BMI and performance variables such as dry matter (DM) intake (DMI) and average daily gain (ADG) compared with BCS [23]. Sheep and goats differ in feed intake, nutrient utilization, and growth performance when consuming condensed tannin-rich diets and forages [27,28]. Consequently, linear measures, BCS, and BMI could differ between these two species and among diets varying in level of tannins. Moreover, meat vs. milk breeds may have different BCS depending on sites of fat deposition. Meat animals deposit considerable amounts of carcass fat, whereas dairy animals deposit proportionally more fat internally relative to greater subcutaneous accretion in meat breeds [29,30,31]. It has been reported that Churra [32] and Finnish Landrace [33] breeds of sheep deposit a higher proportion of noncarcass fat than other breeds. Also, growing animals, depending upon type of animals (i.e., meat or dairy breeds) and diets, tend to develop muscle before they deposit subcutaneous or internal fat [29,30,34,35], which would result in differences of BCS and BMI between the type of animals. It is unclear how BCS, linear measures, and BMI will correlate between themselves and with performances variables of sheep vs. goats selected for different production purposes and with diets varying in quality. It was hypothesized that BMI could correlate better with the performance variables with species differences and linear measures, BCS, and various BMI could be influenced by the dietary treatments and species. Therefore, the objective of this study was to evaluate BCS, linear measures, BMI, and performance variables and their relationships in Alpine doelings (a dairy goat) and Katahdin ewe lambs (meat sheep) fed diets differing in forage type.

## 2. Materials and Methods

### 2.1. Animals, Periods, and Housing

The study was conducted with the approval of the Langston University Animal Care and Use Committee. The duration of the study was 173 days starting in January, 2019. It consisted of four measurement periods, the first three 6 weeks (wk) in length and the last 47 days to obtain data in different times of the growth phases of animals. Twenty-four Alpine doelings (ALP) with an initial BW of 25.3 ± 0.55 kg (mean ± standard error) and age of 10.4 ± 0.11 months and 24 Katahdin ewe lambs (KAT) with initial BW of 28.3 ± 1.02 kg and age of 9.6 ± 0.04 months were utilized. The experiment entailed a 2 × 3 factorial arrangement of treatments, with two species of animals and three dietary treatments. Both ALP and KAT were randomly distributed to the three diets. At most times, animals were housed in six pens (6.1 m × 5.6 m) with a 6.1 m × 1.35 m concrete floor area and a 6.1 m × 4.25 m unpaved floor area in an enclosed building. Each pen had Calan gate feeders (American Calan, Inc., Northwood, NH, USA) for individual feeding. The animals were trained to use of Calan gate feeders for a 2-wk period before the actual start of the experiment. The pens and feeders were aligned in a row adjacent to one another. The same diet was fed to animals of the same species in pens to avoid problems with attempts to gain access to feeders containing a diet other than that assigned due to potential differences in palatability. Animals within pens and treatment groups were randomly assigned to two sets, with set 2 beginning the experiment 1 wk after set 1. This was done because it was not possible to conduct some measures on 48 animals simultaneously.

### 2.2. Diets

Diets were 25% concentrate and 75% coarsely ground forage fed as total mixed diets. Forage was alfalfa hay (ALF), a 1:1 mixture of alfalfa and Sericea lespedeza (*Lespedeza cuneata*) hay (ALF:LES), and lespedeza hay (LES), with a condensed tannin concentration of 0.9, 5.8, and 10.0%, respectively. Concentrate contained rolled corn (19.3%), molasses (5%), and sources of minerals and vitamins (0.7%). Diets averaged 21.2, 17.1, and 13.0% crude protein and 35.3, 39.0, and 40.0% neutral detergent fiber for ALF, ALF+LES, and LES, respectively. Diets were offered at 08:00 h after collecting and weighing refusals, which were approximately 10% of feed intake on the previous few days. In addition, a trace mineral salt block was put in each pen (Big 6 Mineral Salt, American Stockman, Overland Park, KS, USA; containing 965–995 g/kg of NaCl, 4000 mg/kg of Zn, 1600 mg/kg of Fe, 1200 mg/kg of Mn, 260–390 mg/kg of Cu, 100 mg/kg of I, and 40 mg/kg of Co, as fed basis).

### 2.3. Measures

Body weight was measured at the start of the experiment and end of each measurement period (total of 4 periods) to determine ADG in each period and the entire experiment. Performance variables such as DMI, ADG, and the Kleiber ratio (KR) have been reported previously [28]. Body condition score with a scale of 1 to 5 scale and 0.25 increments was determined by three individuals at the start of the experiment and end of each period. The assessment of BCS was described by Ngwa et al. [24] as well as in earlier publications of Villaquiran et al. [36,37]. This method is based on muscle and fat thickness at multiple anatomical locations, including the spinous and transverse processes of the lumbar vertebrate, ribs, and sternum. Linear measures were determined at these times, which were height at the withers (Wither), length from the point of the shoulder to the hook bone (Hook) and pin bone (Pin), circumference from heart girth (Heart), and width at the hook bones (Rump). There were nine BMI calculated as described by Liu et al. [23] and noted below.

BMI-W = BW/Wither [g/cm^2^]

BMI-H= BW/Hook [g/cm^2^]

BMI-P= BW/Pin [g/cm^2^]

BMI-G= BW/Heart [g/cm^2^]

BMI-WH= BW/(Wither × Hook) [g/cm^2^]

BMI-WP= BW/(Wither × Pin) [g/cm^2^]

BMI-GH= BW/(Heart × Hook) [g/cm^2^]

BMI-GP= BW/(Heart × Pin) [g/cm^2^]

BMI-WGH = BW/(Wither × Heart × Hook) [g/cm^3^]

### 2.4. Statistical Analyses

Initial BW, BCS, linear measures, and BMI were analyzed using a mixed effects model with SAS [38] containing fixed effects of species, dietary treatment, and their interactions. Mean separation was through least significant difference with a protected F-test. Pearson correlation coefficients (r) were used to evaluate relationships between variables including the change (∆) in BCS, ADG, DMI, and KR and change (∆) in linear measures and BMI [38]. Canonical Correspondence Analysis (CCA) was also carried out to evaluate relationships between BCS, linear measures, BMI, and performance variables (i.e., BW, ADG, DMI, and KR) using R (version 3.5.1) “CCC” and “Vegan” packages [39] to address BW, BCS, linear measures, and BMI at the end of the study and mean performance variables.

## 3. Results and Discussion

### 3.1. Effects of Species and Dietary Treatment on Change in BW, BCS, Linear Measures, and BMI

Initial BW, BCS, linear measures, and most BMI differed (*p* < 0.05) between KAT and ALP, except for BMI based on length only (i.e., BMI-H and BMI-P; Table 1). Initial BW, BCS, Hook, Pin, Heart, Rump, BMI-W, BMI-WH, and BMI-WP were greater for KAT than for ALP, whereas Wither, BMI-G, BMI-GH, BMI-GP, and BMI-WGH were greater for ALP. The species differences in linear body measurements and consequently BMI agree with findings in a number of other studies [7,10,11,22,40].

There were no significant species × diet interactions in change in values during the experiment (Table 1). Change in Heart and Rump, BCS, and BMI were affected (*p* < 0.05) by species and diet except for BMI-H, BMI-P, BMI-G, BMI-GH, and BMI-WGH. Species differences, when they occurred, typically involved greater values for KAT than for ALP. For diets, values were in most cases greater for ALF than for LES or ALF:LES and LES.

The greater increase in BCS during the experiment for KAT vs. ALP (*p* < 0.05) suggests a greater level of fat in tissues accreted by KAT, which is in addition to the greater initial BCS (*p* < 0.05; Table 1). In the present study, dairy goats and a hair sheep meat breed were used during various stages of growth before maturity. Growing animals bred for meat production tend to deposit more muscle protein and carcass fat rather than high accretion fat in internal depots [22,29,30,34,35,41], which might result in greater BCS in KAT sheep relative to the ALP goat breed for milk production. In the present study, species (sheep and goat) and animal type (meat versus dairy type) were confounded. The species differences between sheep and goat exist for muscle and fat deposition. For example, meat breeds of sheep deposit greater proportions of noncarcass fat and lesser proportions of muscle compared with meat breeds of goats [42]. Similar BCS for ALF and ALF:LES and the lower (*p* < 0.05) value for LES suggests that the composition of tissues accreted by animals consuming diets with ALF was similar and that tissues deposited by animals ingesting LES was higher in protein and lower in fat, in addition to less tissue gain. However, it is realized that BCS is subjective and determined largely by the amount and distribution of subcutaneous fat. The deposition of fat in internal sites may have influenced the relationship between BCS and variables such as ADG [22,23,32].

It has been suggested that Heart alone or combined with a measure of length or height could be used to accurately predict BW [16]. Relatedly, feed restriction lowered BMI based on Wither and body length [26]. Palm oil or rapeseed oil fed to lactating goats caused differences in BW and BMI based on height and body length but not BCS, likely due to internal body fat mobilization to maintain milk production [22]. In regard to the discussion above, it would be useful to evaluate relationships between these variables to identify most appropriate linear measures and BMI for use rather than only viewing breed or species means.

### 3.2. Relationships between BCS, Linear Measures, and BMI

There were significant moderate to high correlations (*p* < 0.001) between BCS and all linear measures for both species, except for the correlation coefficient of 0.18 (*p* = 0.047) between BCS and Rump for ALP (Table 2). The correlations were always greater for KAT than for ALP, except for similar values between Heart and Pin. Among the correlations between BCS and linear measures, the greatest value was between BCS and Pin for ALP and between BCS and Heart for KAT. Liu et al. [23] reported that BCS of Alpine doelings had positive low to moderate correlations with all linear measures except Wither, with the greatest relationship for Heart. In the present experiment, without considering correlation between Hook and Pin, which would be expected to be high relative to other measures, the greatest association was observed between Heart and Pin (0.71), followed by Heart and Hook (0.66). For KAT, the highest correlation was between Heart and Rump (0.82), followed by similar values for Heart-Wither, Heart-Hook, and Hook-Rump (0.68 to 0.69). These findings are in accordance with those of Liu et al. [23] as well.

There were positive correlations (*p* < 0.001) between BCS and all BMI for both species, with greater values for KAT than for ALP (Table 3). From these results, as would be expected, it is evident that BMI were better predictors of BCS than linear measures alone. Although BCS had the greatest correlation with Pin for ALP and with Heart for KAT, the BMI calculated from these linear measures were not more highly related to BCS compared with other BMI. This type of discrepancy was also noted by Liu et al. [23]. There were relatively small differences among BMI in regard to the relationship with BCS, with correlations for ALP ranging from 0.63 to 0.71 (excluding the r of 0.52 for BMI-P) and for KAT ranging from 0.77 to 0.86. Ptaček al. [4], Chavarria-Aguilar et al. [43], and Eknæs et al. [22] used similar BMI to assess animal conditions. In addition to differences between species such as in the present experiment, breed within species could have impact. For example, the correlation between BCS and BMI based on height and length measures was stronger (r = 0.80) in Pelibuey ewes [43] compared with the relationship between BCS and a BMI also calculated from height and length (r = 0.39) in Wallachian sheep [22]. 

All correlations between BMI were highly significant (*p* < 0.001) in both species. In ALP, the lowest association was for BMI-P with BMI-G (0.61) and the highest was between BMI-WH and BMI-WGH and between BMI-GH and BMI-GP (0.93). In KAT, the lowest correlation was also between BMI-P and BMI-G (0.75) and the highest was between BMI-GH and BMI-GP (0.97), with values higher than for ALP. 

### 3.3. Relationships for BCS and Linear Measures with Performance

Diet and species effects on DMI, ADG, and KR were addressed by Wang et al. [28]. There were no species × diet interactions in variables addressed in this report. Dry matter intake tended (*p* = 0.063) to be greater for KAT than for ALP (4.14 and 3.84% BW, respectively) and was similar among diets. Average daily gain (180 and 88 g) and KR (10.5 and 6.3 g/kg BW^0.75^, respectively) were greater for KAT than for ALP. Diet had a marked effect on ADG, with differences among each and values of 159, 132, and 111 g/day for ALF, ALF:LES, and LES, respectively, in accordance with change in BW. Factors that appeared responsible for differences in ADG among diets, despite similar feed intake, appeared to be dietary levels of condensed tannins (CT; 0.9, 5.8, and 10.0%) and crude protein (21.2, 17.1, and 13.0% for ALF, ALF:LES, and LES, respectively). Digestibility decreased with increasing levels of lespedeza and CT, particularly of N. Wang et al. [28] postulated that decreasing amino acid absorption and protein status with increasing level of lespedeza may have prevented an increase in feed intake that has been observed in some other studies [44,45].

For ALP, there was a correlation (*p* < 0.05) between BCS and BW, but the correlations between BCS and ADG, DMI in g/day and relative to BW and kg BW^0.75^, and KR were not significant (Table 4). However, *p* values for correlation coefficients between BCS and ADG (r = 0.38, *p* = 0.065) and DMI in g/day (r = 0.35; *p* = 0.093) only approached significance. For KAT, BCS was, however, correlated with BW (*p* < 0.001), ADG (*p* < 0.001), and DMI in g/day (*p*<0.001), with moderate to high correlations. Likewise, Delfa et al. [20] found that BCS was correlated with BW of goats. Relationships between BCS and milk production have been reported in Scottish Halfbred [46] and Awassi [47] goat breeds as a result of mobilization of body fat and protein reserves in early lactation. In a study with ALP doelings fed high forage diets, low to moderate relationships (r = 0.21 to 0.39) were noted between BCS and BW, DMI in g/day, and ADG [23]. All of these results indicate that BCS can be an indicator or reflective of animal performance, but the strength of such relationships will depend on many factors. For example, the much higher association between BCS and BW for KAT than for ALP could be attributed to differences in sites of fat deposition, with potentially relatively more subcutaneous and less internal fat depots in KAT vs. ALP [3,32]. Similarly, it has been suggested that Churra [32] and Finnish Landrace [33] breeds of sheep deposit a greater proportion of noncarcass fat than other breeds. The magnitude of change in BW per unit of BCS differs among breeds depending on body size, conformation, fat distribution throughout the body, and standard reference body weight [3,33,48]. The stronger relationships between BCS and production variables for KAT compared with ALP in the present study are unclear, but may involve differences in BCS scoring in the two species. Katahdin has been selected for meat production, which would include deposition of considerable carcass fat, whereas ALP is a dairy or milk breed that deposits considerable fat internally relative to greater subcutaneous accretion in meat goat breeds [29,30], similar to differences between dairy and beef cattle breeds [31].

Body weight was correlated (*p* < 0.05) with all linear measures, with the highest correlation for Pin (0.85) and lowest value for Wither (0.48; Table 4). For KAT, BW also was correlated (*p* < 0.05) with all linear measures, with the lowest coefficient for Wither (0.68) and highest value for Rump (0.91). High correlations between linear measures and BW have been observed in many other studies as well (i.e., [1,18,23,49]).

No linear measure for ALP was correlated with ADG, DMI in g/day, % BW, or g/kg BW^0.75^, or KR (Table 4). In contrast, for KAT, DMI in g/day was correlated (*p* < 0.05) with all linear measures (r of 0.41 to 0.70), with the greatest value for Rump (0.70), followed by Hook (0.63). Moreover, ADG of KAT was correlated (*p* < 0.05) with Heart and Rump. However, similar to ALP, other variables for KAT such as DMI in % BW and g/kg BW^0.75^ and KR were not related to linear measures. Liu et al. [23] reported that correlations for ALP doelings between linear measures, except for Pin and DMI in g/day and g/kg BW^0.75^ and ADG, were of low to moderate magnitude. The highest correlation coefficients were for Heart and DMI in g/day (0.78) and % BW (0.53), whereas for DMI in g/kg BW^0.75^, the coefficients of 0.42–0.49 were noted for Wither, Hook, Heart, and Rump.

### 3.4. Relationships between BMI and Performance

From the CCA biplot for ALP, it was noted that BMI were highly related to variables of BW, ADG, DMI in g/day, and KR, whereas BCS was not related to any performance variable (Figure 1). In KAT also, ADG, BW, and DMI in g/day were highly related to BMI-WH, as was also true between KR and BMI-W and BMI-WP, but there were no relationships between BCS and performance variables (Figure 2). This likely involves the fact that BW is used in estimating both BMI and these expressions of DMI.

As expected based on how BMI are calculated, BW of both species was related (*p* < 0.05) to all BMI (Table 5). For ALP, correlation coefficients ranged from 0.58 for BMI-H to 0.88 for BMI-WP and for KAT from 0.48 for BMI-WGH to 0.91 for BMI-GH. These correlations compare with those for BCS of ALP and KAT of 0.49 and 0.91, respectively. The correlations between BW and BMI based on combinations of two linear measures were in most cases greater than the correlation between BMI calculated from one or three linear measures. Likewise, Liu et al. [23] noted that associations between BW and a BMI based on Wither, Heart, and Pin were not significant and, thus, this BMI was not recommended. Similar to BW, ADG were correlated (*p* < 0.05) with all BMI of both species, with correlation coefficients for ALP ranging from 0.41 for BMI-W to 0.65 for BMI-GH and for KAT from 0.47 for BMI-G to 0.72 for BMI-GH. The associations between ADG and BMI were always greater for KAT than for ALP, except for BMI-WGH, with a lower value for KAT. Overall, BMI were more highly related to ADG than were linear measures. It is also important to note that BCS of ALP was not related to ADG, and the correlation coefficient for KAT was 0.57 and less than for most BMI. In Wallachian sheep, milk production was significantly correlated with BMI calculated from height and length [4]. Similarly, goats with a relatively high BMI produced more milk than those with low BMI [50]. 

Dry matter intake in g/day by ALP was correlated (*p* < 0.05) with five of the nine BMI, BMI-P, BMI-G, BMI-WP, BMI-GH and BMI-GP, with the highest (0.61) for BMI-GP and lowest (0.43) for BMI-WP. Dry matter intake by KAT in g/day was correlated with all BMI, with the highest correlation for BMI-GH and BMI-GP (0.75) and lowest for BMI-WGH (0.52), and all correlations were greater for KAT than for ALP. Similar to ADG, BCS of ALP was not significantly related to DMI in g/day, and the correlations for KAT were less than for BMI based on two linear measures (i.e., 0.63). Liu et al. [23] reported that there were significant relations between DMI by ALP doelings in g/day and all BMI, except for BMI-WGH, which is in accordance with results of the present study for ALP but not KAT. Liu et al. [23] also noted that the relation between DMI in g/day and BMI was greatest for BMI-WP (0.63), BMI-GP (0.58), and BMI-P (0.56). In the present experiment, DMI in % BW and g/kg BW^0.75^ was not related (*p* > 0.05) to BMI for either species, and the same was true for BCS. As noted before, this could relate to influence of BW on both variables. In contrast, Liu et al. [23] noted significant correlation between DMI in % BW and g/kg BW^0.75^ by ALP doelings and some BMI, with greatest values for BMI-P and BMI-WP. Interestingly, and in contrast with involvement of BW in estimation of BMI and DMI in % BW and g/kg BW^0.75^, the KR of ALP was related to some BMI, though the correlations were moderate (i.e., 0.42, 0.41, and 0.43 for BMI-H, BMI-GH, and BMI-WGH, respectively). However, the KR of KAT was not related to any BMI, and BCS of both species was not related to KR. 

### 3.5. Relationships between Changes (∆) in Performance Variables and BCS and BMI

There were few significant correlations between ∆BCS and change in linear measures and ∆BMI (Table 6). For ALP, ∆BCS was only correlated with ∆Wither (r = 0.45). The BCS of KAT was correlated (*p* < 0.05) with ∆BMI-H, ∆BMI-WH, ∆BMI-WP, and ∆BMI-GH (r = 0.47, 0.52, 0.42, and 0.42, respectively). There were correlations (*p* < 0.05) between ADG and change in all linear measures except ∆Wither for both ALP and KAT, suggesting that ∆ADG is not highly associated with proportional height changes. For ALP, ADG was correlated (*p* < 0.05) with all ∆BMI except ∆BMI-*p* and ∆BMI-H, with the greatest correlation for ∆BMI-WH (0.77) and ∆BMI-W (0.76). Significant correlations for KAT were noted for ∆BMI-W, ∆BMI-P, ∆BMI-WH, ∆BMI-WP, and ∆BMI-GP, with the greatest value for ∆BMI-W (0.83), followed by ∆BMI-WP (0.75). Although ∆Wither was not related to ∆ADG, ∆BMI based on Wither was highly related to ∆ADG for both species. The correlation between ∆BCS for ALP and ADG was not significant, and that for KAT was significant but lower than for ∆BMI-W, ∆BMI-WH, and ∆BMI-WP. Interestingly, change in KR was correlated (*p* < 0.05) with the same variables for which there were correlations with ADG for both species. No significant correlations were noted between any expressions of ∆DMI with changes in any linear measures or ∆BMI for ALP. For KAT, ∆DMI in g/day, % BW, and g/kg BW^0.75^ were correlated (*p* < 0.05) with ∆Heart but not with change in other linear measures. For KAT, ∆DMI in g/day was related (*p* < 0.05) to ∆BMI-W, ∆BMI-WH, and ∆BMI-WP (r = 0.50, 0.48, and 0.49, respectively). For KAT, there was no significant correlation for DMI in % BW for either species involving change in BMI. For KAT, DMI in g/kg BW^0.75^ was moderately related (*p* < 0.05) only to two ∆BMI variables, i.e., ∆BMI-P and ∆BMI-WP (0.44 and 0.43, respectively). For both species, ∆BCS was not correlated (*p* > 0.05) with any DMI expressions.

## 4. Summary and Conclusions

We hypothesized that the changes in linear measures, BCS, and various BMI may be affected by species because of factors such as dietary preferences. However, these variables were not affected by any species × diet interactions, but were mostly affected by species and diet. All linear measures were positively correlated with BCS, BW, or BMI in both breeds, which were usually greater for KAT than for ALP. Body condition score was positively correlated with BW, DMI (g/day), and ADG of KAT and tended to be correlated for ALP as well. Linear measures were not correlated with ADG or DMI in ALP, but all linear measures were correlated with DMI, and Pin and Rump were related to ADG of KAT. All BMI had positive correlations with BW and ADG in both species, with the highest values generally for BMI-GH or BMI-GP, which were usually greater for KAT than for ALP. Among the BMI, DMI had the highest positive correlation with BMI-GP in ALP and BMI-GH or BMI-GP in KAT. Taken together, diets influenced changes in all linear measures and some BMI and there were species differences in the correlations between linear measures, BCS, and BMI vs. performance traits. Body mass indexes based on body length and height or heart girth can be recommended as preferred management tools relative to BCS because of less subjectivity and stronger relationships with performance measures, although BCS may provide an assessment of subcutaneous fatness and carcass composition not imparted by BMI that relate more to whole body composition. In the present study, species (goats and sheep) and animal type (meat and dairy animals) were confounded. Further studies using similar animal types with different species may be performed to understand the relationship among the BCS, BMI and performance measures.

## Figures and Tables

**Figure 1 animals-12-03183-f001:**
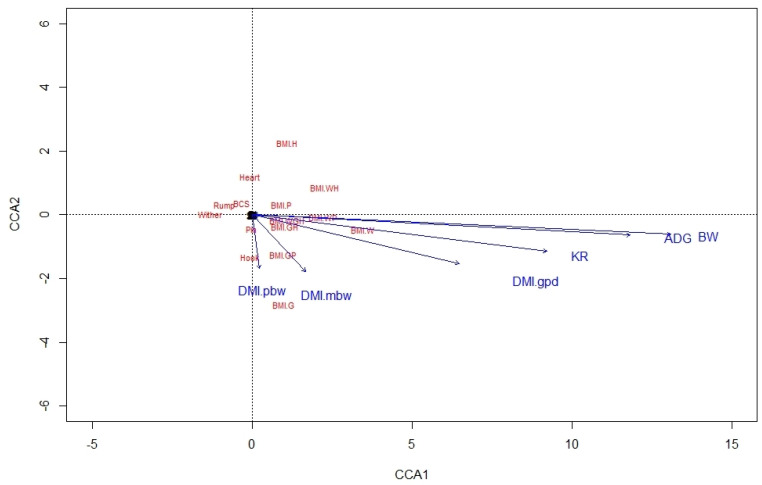
A canonical correspondence analysis (CCA) displaying the relationship between the linear body measures, body condition score, and body mass indexes and performance variables for Alpine doelings. Arrows show the direction of the gradient, with stronger relationships depicted by longer arrows. BCS = body condition score; BW = body weight; ADG = average daily BW gain; DMI.gpd, DMI.pbw and DMI.mbw = dry matter intake in gram per day, percent of BW, and gram per kilogram metabolic BW, respectively; KR = Kleiber ratio (g/kg BW^0.75^); Wither = height at withers; Hook = point of the shoulder to hook bone; Pin = point of the shoulder to pin bone; Heart = heart girth; Rump = width at hook bones; BMI = body mass index; BMI.W = BW/Wither [g/cm^2^]; BMI.H = BW/Hook [g/cm^2^]; BMI.P = BW/Pin [g/cm^2^]; BMI.G = BW/Heart [g/cm^2^]; BMI.WH = BW/(Wither × Hook) [g/cm^2^]; BMI.WP = BW/(Wither × Pin) [g/cm^2^]; BMI.GH = BW/(Heart × Hook) [g/cm^2^]; BMI.GP = BW/(Heart × Pin) [g/cm^2^]; BMI.WGH = BW/(Wither × Heart × Hook) [g/cm^3^].

**Figure 2 animals-12-03183-f002:**
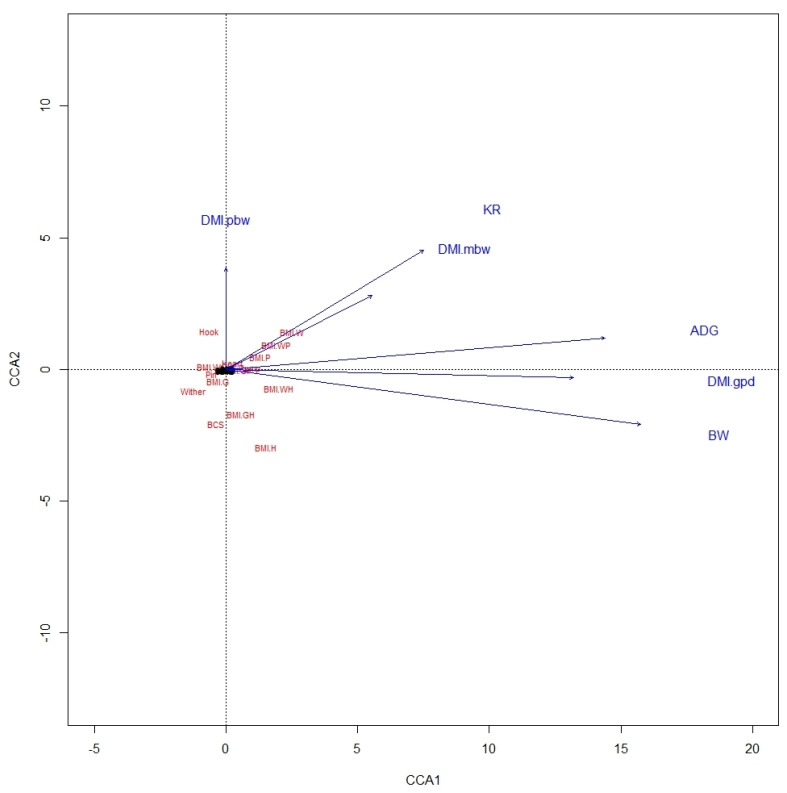
A canonical correspondence analysis (CCA) displaying the relationship between the linear body measures, body condition score, and body mass indexes and performance variables of Katahdin ewe lambs. Arrows show the direction of the gradient, with stronger relationships depicted by longer arrows. The abbreviations are defined in Figure 1.

**Table 1 animals-12-03183-t001:** The effects of species and diet on initial body weight, body condition score, linear measures, and body mass indexes and change during the experiment ^1^.

	*p* Value ^2^			Species			Diet			
Item ^3^	Species	Diet	Species × Diet	ALP	KAT	SEM	ALF	ALF:LES	LES	SEM
Initial values										
BW	0.013			25.3 ^a^	28.3 ^b^	0.82				
BCS	<0.001			2.74 ^a^	3.01 ^b^	0.037				
Hook	0.002			51.3 ^a^	54.8 ^b^	0.73				
Pin	<0.001			62.5 ^a^	67.5 ^b^	0.80				
Heart	<0.001			71.8 ^a^	80.9 ^b^	1.02				
Wither	0.028			63.3 ^b^	61.2 ^a^	0.66				
Rump	0.001			14.2 ^a^	15.8 ^b^	0.29				
BMI-W	<0.001			6.31 ^a^	7.53 ^b^	0.171				
BMI-H	0.544			9.59	9.41	0.544				
BMI-P	0.147			6.46	6.18	0.133				
BMI-G	<0.001			4.91 ^b^	4.30 ^a^	0.085				
BMI-WH	0.008			7.77 ^a^	8.39 ^b^	0.158				
BMI-WP	0.021			6.38 ^a^	6.81 ^b^	0.127				
BMI-GH	0.001			6.85 ^b^	6.34 ^a^	0.105				
BMI-GP	<0.001			5.62 ^b^	5.15 ^a^	0.084				
BMI-WGH	0.045			0.108 ^b^	0.104 ^a^	0.0015				
Change										
BW	<0.001	<0.001	0.114	15.1 ^a^	31.0 ^b^	0.63	27.4 ^c^	22.6 ^b^	19.0 ^a^	0.77
BCS	<0.001	0.019	0.143	0.31 ^a^	0.66 ^b^	0.045	0.57 ^b^	0.53 ^b^	0.35 ^a^	0.055
Hook	0.333	0.003	0.233	6.5	7.5	0.72	9.6 ^b^	6.4 ^a^	5.2 ^a^	0.88
Pin	0.484	<0.001	0.316	10.1	10.8	0.67	13.3 ^b^	9.4 ^a^	8.6 ^a^	0.82
Heart	<0.001	<0.001	0.085	8.5 ^a^	13.0 ^b^	0.72	13.7 ^c^	10.8 ^b^	7.8 ^a^	0.88
Wither	0.056	0.043	0.527	4.7	6.0	0.48	6.0 ^b^	6.0 ^b^	4.1 ^a^	0.59
Rump	0.001	0.024	0.062	1.1 ^a^	2.5 ^b^	0.26	2.5 ^b^	1.7 ^ab^	1.3 ^a^	0.31
BMI-W	<0.001	0.001	0.627	2.41 ^a^	5.56 ^b^	0.183	4.73 ^b^	3.81 ^a^	3.41 ^a^	0.224
BMI-H	<0.001	0.362	0.454	2.44 ^a^	5.84 ^b^	0.271	4.34	4.32	3.75	0.332
BMI-P	<0.001	0.101	0.555	1.17 ^a^	3.49 ^b^	0.155	2.52	2.47	1.99	0.190
BMI-G	<0.001	0.120	0.067	1.35 ^a^	2.39 ^b^	0.073	2.02	1.84	1.76	0.089
BMI-WH	<0.001	0.005	0.456	2.46 ^a^	5.71 ^b^	0.168	4.61 ^b^	4.07 ^ab^	3.59 ^a^	0.206
BMI-WP	<0.001	0.003	0.476	1.77 ^a^	4.43 ^b^	0.140	3.55 ^b^	3.10 ^ab^	2.65 ^a^	0.172
BMI-GH	<0.001	0.142	0.566	1.82 ^a^	3.74 ^b^	0.109	2.95	2.82	2.57	0.134
BMI-GP	<0.001	0.042	0.882	1.28 ^a^	2.89 ^b^	0.083	2.25 ^b^	2.12 ^ab^	1.88 ^a^	0.101
BMI-WGH	<0.001	0.836	0.265	0.019 ^a^	0.046 ^b^	0.0018	0.024	0.032	0.032	0.0022

^1^ Species were Alpine (ALP) doelings and Katahdin (KAT) ewe lambs; diets were 75% forage consisting of alfalfa hay (ALF), a 1:1 mixture of alfalfa and Sericea lespedeza hay (ALF:LES), and lespedeza hay (LES); periods 1–3 were 42 days in length and period 4 was 47 days, for a total of 173 days; measures occurred at the beginning and end of the experiment. ^2^ BW = body weight (kg); ^3^ BCS = body condition score (1–5); Hook = point of the shoulder to hook bone (cm); Pin = point of the shoulder to pin bone (cm); Heart = heart girth (cm); Wither = height at withers (cm); Rump = width at hook bones (cm); BMI = body mass index; BMI-W = BW/Wither [g/cm^2^]; BMI-H = BW/Hook [g/cm^2^]; BMI-P = BW/Pin [g/cm^2^]; BMI-G = BW/Heart [g/cm^2^]; BMI-WH = BW/(Wither × Hook) [g/cm^2^]; BMI-WP = BW/(Wither × Pin) [g/cm^2^]; BMI-GH = BW/(Heart × Hook) [g/cm^2^]; BMI-GP = BW/(Heart × Pin) [g/cm^2^]; BMI-WGH = BW/(Wither × Heart × Hook) [g/cm^3^]; ^a,b,c^ Means within a grouping without a common superscript letter differ (*p* < 0.05).

**Table 2 animals-12-03183-t002:** Pearson correlation coefficients (r) between body condition score and linear measures ^1^.

		Variable ^2^				
Species	Item ^3^	Wither	Hook	Pin	Heart	Rump
Alpine	BCS	0.40	0.37	0.55	0.49	0.18
	Wither		0.58	0.61	0.54	0.48
	Hook			0.81	0.66	0.54
	Pin				0.71	0.49
	Heart					0.59
Katahdin	BCS	0.64	0.65	0.70	0.73	0.57
	Wither		0.62	0.62	0.69	0.63
	Hook			0.87	0.68	0.68
	Pin				0.68	0.65
	Heart					0.82

^1^ Based on individual measurements at the beginning of the experiment and end of the four periods; *p* values were <0.001 except for a *p* of 0.047 for the r for Alpine between BCS and Rump. ^2^ Wither = height at withers; Hook = point of the shoulder to hook bone; Pin = point of the shoulder to pin bone; Heart = heart girth; Rump = width at hook bones. ^3^ BCS = body condition score (1–5).

**Table 3 animals-12-03183-t003:** Pearson correlation coefficients (r) between body condition score and body mass indexes ^1^.

		Variable ^2^								
Species	Item ^3^	BMI-W	BMI-H	BMI-P	BMI-G	BMI-WH	BMI-WP	BMI-GH	BMI-GP	BMI-WGH
Alpine	BCS	0.63	0.68	0.52	0.65	0.70	0.63	0.73	0.66	0.71
	BMI-W		0.71	0.68	0.75	0.94	0.94	0.80	0.80	0.85
	BMI-H			0.76	0.66	0.91	0.79	0.91	0.79	0.87
	BMI-P				0.61	0.77	0.88	0.75	0.88	0.68
	BMI-G					0.77	0.75	0.90	0.91	0.79
	BMI-WH						0.95	0.92	0.86	0.93
	BMI-WP							0.85	0.91	0.85
	BMI-GH								0.93	0.91
	BMI-GP									0.82
Katahdin	BCS	0.81	0.83	0.80	0.77	0.86	0.85	0.85	0.84	0.77
	BMI-W		0.84	0.83	0.84	0.96	0.96	0.89	0.89	0.91
	BMI-H			0.92	0.77	0.95	0.92	0.95	0.91	0.88
	BMI-P				0.75	0.91	0.95	0.89	0.94	0.81
	BMI-G					0.84	0.84	0.94	0.93	0.90
	BMI-WH						0.98	0.96	0.94	0.93
	BMI-WP							0.93	0.96	0.90
	BMI-GH								0.97	0.95
	BMI-GP									0.91

^1^ Based on individual measurements at the beginning of the experiment and end of the four periods; all *p* values were <0.001. ^2^ BMI = body mass index; BMI-W = BW/Wither [g/cm^2^]; BMI-H = BW/Hook [g/cm^2^]; BMI-P = BW/Pin [g/cm^2^]; BMI-G = BW/Heart [g/cm^2^]; BMI-WH = BW/(Wither × Hook) [g/cm^2^]; BMI-WP = BW/(Wither × Pin) [g/cm^2^]; BMI-GH = BW/(Heart × Hook) [g/cm^2^]; BMI-GP = BW/(Heart × Pin) [g/cm^2^]; BMI-WGP = BW/(Wither × Heart × Hook) [g/cm^3^]. ^3^ BCS = body condition score (1–5).

**Table 4 animals-12-03183-t004:** Pearson correlation coefficients (r) between body weight, average daily gain, dry matter intake, Kleiber ratio, body condition score, and linear measures ^1^.

			Variable ^2^					
Species	Item ^3^	Estimate	BCS	Wither	Hook	Pin	Heart	Rump
Alpine	BW (kg)	r	0.49	0.48	0.76	0.85	0.77	0.59
		*p*	0.015	0.017	<0.001	<0.001	<0.001	0.003
	ADG (g)	r	0.38	0.22	0.11	0.35	0.24	0.10
		*p*	0.065	0.294	0.616	0.097	0.260	0.645
	DMI (g/day)	r	0.35	0.29	0.30	0.17	0.15	0.15
		*p*	0.093	0.164	0.155	0.418	0.496	0.489
	DMI (% BW)	r	0.05	0.04	−0.16	−0.35	−0.35	−0.19
		*p*	0.815	0.840	0.447	0.092	0.097	0.379
	DMI (g/kg BW^0.75^)	r	0.13	0.12	−0.04	−0.22	−0.22	−0.10
		*p*	0.542	0.588	0.856	0.302	0.292	0.637
	Kleiber ratio (g/kg BW^0.75^)	r	0.23	0.03	−0.18	0.05	−0.07	−0.15
		*p*	0.263	0.880	0.399	0.810	0.743	0.492
Katahdin	BW	r	0.91	0.68	0.83	0.82	0.87	0.91
		*p*	<0.001	<0.001	<0.001	<0.001	<0.001	<0.001
	ADG	r	0.57	0.31	0.36	0.35	0.50	0.50
		*p*	0.004	0.141	0.087	0.091	0.014	0.014
	DMI (g/day)	r	0.63	0.41	0.63	0.61	0.60	0.70
		*p*	<0.001	0.049	0.001	0.001	0.002	<0.001
	DMI (% BW)	r	−0.33	−0.39	−0.26	−0.27	−0.33	−0.23
		*p*	0.110	0.058	0.211	0.211	0.112	0.274
	DMI (g/kg BW^0.75^)	r	−0.001	−0.13	0.07	0.06	0.00	0.11
		*p*	0.998	0.545	0.741	0.791	0.983	0.61
	Kleiber ratio (g/kg BW^0.75^)	r	−0.01	−0.17	−0.23	−0.21	−0.07	−0.10
		*p*	0.976	0.425	0.278	0.321	0.762	0.645

^1^ Based on the one value for the entire experiment or average of measures taken at the beginning of the study and end of the 4 periods. ^2^ BCS = body condition score (1–5); Wither = height at withers; Hook = point of the shoulder to hook bone; Pin = point of the shoulder to pin bone; Heart = heart girth; Rump = width at hook bones. ^3^ BW = body weigh; ADG = average daily gain; DMI = dry matter intake.

**Table 5 animals-12-03183-t005:** Pearson correlation coefficients (r) between body mass indexes and body weight, average daily gain, dry matter intake, and Kleiber ratio ^1^.

			Variable ^2^					
Species	Item ^3^	Estimate	BW	ADG	DMI (g/day)	DMI (% BW)	DMI (g/kg BW^0.75^)	KR
Alpine	BMI-W	r	0.85	0.41	0.32	−0.22	−0.08	0.12
		*p*	<0.001	0.045	0.130	0.299	0.698	0.575
	BMI-H	r	0.58	0.60	0.25	−0.09	−0.01	0.42
		*p*	0.003	0.002	0.234	0.660	0.974	0.042
	BMI-p	r	0.71	0.42	0.52	0.10	0.22	0.16
		*p*	<0.001	0.039	0.009	0.629	0.302	0.452
	BMI-G	r	0.70	0.46	0.48	0.09	0.20	0.25
		*p*	<0.001	0.024	0.018	0.681	0.352	0.241
	BMI-WH	r	0.84	0.57	0.33	−0.19	−0.06	0.29
		*p*	<0.001	0.004	0.118	0.362	0.774	0.178
	BMI-WP	r	0.88	0.46	0.43	−0.11	0.03	0.15
		*p*	<0.001	0.025	0.037	0.602	0.891	0.491
	BMI-GH	r	0.78	0.65	0.44	−0.01	0.12	0.41
		*p*	<0.001	0.001	0.029	0.982	0.591	0.044
	BMI-GP	r	0.85	0.54	0.61	0.12	0.26	0.26
		*p*	<0.001	0.006	0.002	0.572	0.223	0.224
	BMI-WGH	r	0.59	0.59	0.34	−0.01	0.08	0.43
		*p*	0.003	0.003	0.104	0.949	0.716	0.035
Katahdin	BMI-W	r	0.81	0.59	0.70	−0.04	0.25	0.12
		*p*	<0.001	0.003	<0.001	0.837	0.230	0.568
	BMI-H	r	0.79	0.66	0.60	−0.16	0.12	0.24
		*p*	<0.001	<0.001	0.002	0.453	0.588	0.262
	BMI-P	r	0.77	0.66	0.60	−0.13	0.14	0.24
		*p*	<0.001	<0.001	<0.001	0.541	0.515	0.258
	BMI-G	r	0.66	0.47	0.61	−0.02	0.23	0.06
		*p*	<0.001	0.021	0.002	0.912	0.284	0.777
	BMI-WH	r	0.88	0.68	0.72	−0.10	0.21	0.19
		*p*	<0.001	<0.001	<0.001	0.630	0.318	0.376
	BMI-WP	r	0.88	0.70	0.74	−0.09	0.23	0.20
		*p*	<0.001	<0.001	<0.001	0.667	0.284	0.352
	BMI-GH	r	0.91	0.72	0.75	−0.10	0.22	0.20
		*p*	<0.001	<0.001	<0.001	0.571	0.326	0.353
	BMI-GP	r	0.89	0.71	0.75	−0.10	0.22	0.20
		*p*	<0.001	<0.001	<0.001	0.639	0.294	0.351
	BMI-WGH	r	0.48	0.54	0.52	0.17	0.33	0.34
		*p*	0.016	0.004	0.007	0.369	0.086	0.091

^1^ Based on the one value for the entire experiment or average of measures at the beginning of the study and end of the 4 periods. ^2^ BW = body weight (kg); ADG = average daily gain (g); DMI = dry matter intake; KR = Kleiber ratio (g/kg BW^0.75^). ^3^ BMI = body mass index; BMI-W = BW/Wither [g/cm^2^]; BMI-H = BW/Hook [g/cm^2^]; BMI-P = BW/Pin [g/cm^2^]; BMI-G = BW/Heart [g/cm^2^]; BMI-WH = BW/(Wither × Hook) [g/cm^2^]; BMI-WP = BW/(Wither × Pin) [g/cm^2^]; BMI-GH = BW/(Heart × Hook) [g/cm^2^]; BMI-GP = BW/(Heart × Pin) [g/cm^2^]; BMI-WGP = BW/(Wither × Heart × Hook) [g/cm^3^].

**Table 6 animals-12-03183-t006:** Pearson correlation coefficients (r) between change (∆) in body condition score, average daily gain, dry matter intake, and Kleiber ratio and change (∆) in linear measures and body mass indexes during the experiment.

			Variable ^1^					
Species	Item ^2^	Estimate	∆BCS	ADG	DMI (g/day)	DMI (% BW)	DMI (g/kg BW^0.75^)	KR
Alpine	∆Wither	r	0.45	0.28	−0.21	−0.24	−0.25	0.31
		*p*	0.028	0.178	0.303	0.250	0.246	0.143
	∆Hook	r	0.01	0.57	0.19	0.26	0.25	0.69
		*p*	0.962	0.003	0.372	0.215	0.239	<0.001
	∆Pin	r	0.20	0.53	0.11	0.10	0.10	0.58
		*p*	0.348	0.008	0.618	0.638	0.631	0.003
	∆Heart	r	0.05	0.52	0.17	0.16	0.17	0.56
		*p*	0.814	0.009	0.426	0.452	0.440	0.004
	∆Rump	r	−0.17	0.53	−0.01	−0.06	−0.05	0.57
		*p*	0.422	0.007	0.996	0.773	0.811	0.004
	∆BMI-W	r	−0.31	0.76	0.35	0.23	0.27	0.76
		*p*	0.144	<0.001	0.091	0.278	0.208	<0.001
	∆BMI-H	r	0.04	0.39	−0.02	−0.20	−0.16	0.31
		*p*	0.860	0.057	0.929	0.346	0.450	0.138
	∆BMI-P	r	−0.15	0.39	0.07	0.01	0.02	0.39
		*p*	0.283	0.058	0.730	0.969	0.910	0.058
	∆BMI-G	r	0.05	0.63	0.05	−0.03	−0.01	0.66
		*p*	0.814	0.001	0.807	0.886	0.956	0.001
	∆BMI-WH	r	−0.19	0.77	0.23	0.05	0.09	0.73
		*p*	0.369	<0.001	0.276	0.821	0.660	<0.001
	∆BMI-WP	r	−0.28	0.71	0.26	0.16	0.19	0.71
		*p*	0.178	<0.001	0.212	0.458	0.377	<0.001
	∆BMI-GH	r	0.06	0.70	0.02	−0.15	−0.12	0.66
		*p*	0.788	<0.001	0.690	0.954	0.960	<0.001
	∆BMI-GP	r	−0.08	0.68	0.09	−0.01	0.01	0.69
		*p*	0.698	<0.001	0.689	0.954	0.960	<0.001
	∆BMI-WGH	r	−0.23	0.52	0.13	0.00	0.03	0.50
		*p*	0.272	0.009	0.541	0.995	0.887	0.014
	∆BCS	r		0.07	0.10	0.11	0.15	0.07
		*p*		0.738	0.643	0.464	0.486	0.733
Katahdin	∆Wither	r	0.15	0.02	−0.14	0.31	0.18	0.31
		*p*	0.477	0.923	0.510	0.139	0.411	0.140
	∆Hook	r	−0.07	0.41	0.08	0.29	0.24	0.61
		*p*	0.737	0.048	0.716	0.171	0.261	0.002
	∆Pin	r	−0.02	0.42	0.03	0.07	0.06	0.54
		*p*	0.919	0.040	0.888	0.744	0.797	0.006
	∆Heart	r	0.34	0.71	0.42	0.56	0.58	0.84
		*p*	0.100	<0.001	0.042	0.005	0.003	<0.001
	∆Rump	r	−0.14	0.49	0.15	0.26	0.25	0.63
		*p*	0.508	0.014	0.478	0.223	0.237	<0.001
	∆BMI-W	r	0.33	0.83	0.50	0.18	0.33	0.72
		*p*	0.117	<0.001	0.013	0.387	0.115	0.001
	∆BMI-H	r	0.471	0.283	0.26	0.00	0.10	0.13
		*p*	0.020	0.180	0.219	0.997	0.626	0.560
	∆BMI-P	r	0.40	0.47	0.36	0.39	0.44	0.48
		*p*	0.051	0.022	0.084	0.058	0.033	0.017
	∆BMI-G	r	0.01	0.12	−0.18	−0.23	−0.25	0.16
		*p*	0.966	0.575	0.409	0.296	0.236	0.466
	∆BMI-WH	r	0.52	0.70	0.48	0.11	0.27	0.54
		*p*	0.010	<0.001	0.019	0.594	0.199	0.007
	∆BMI-WP	r	0.42	0.75	0.49	0.32	0.43	0.70
		*p*	0.040	<0.001	0.015	0.124	0.035	<0.001
	∆BMI-GH	r	0.42	0.30	0.16	−0.09	−0.01	0.18
		*p*	0.043	0.155	0.468	0.684	0.971	0.402
	∆BMI-GP	r	0.35	0.45	0.23	0.23	0.25	0.47
		*p*	0.092	0.029	0.288	0.287	0.231	0.019
	∆BMI-WGH	r	0.25	0.16	0.08	−0.16	−0.09	0.03
		*p*	0.240	0.458	0.697	0.442	0.662	0.900
	∆BCS	r		0.53	0.37	0.02	0.15	0.34
		*p*		0.008	0.077	0.941	0.472	0.099

^1^ BCS = change in body condition score; DMI = DM intake; KR = Kleiber ratio (g/kg BW^0.75^).^2^ Wither = height at withers; Hook = point of the shoulder to hook bone; Pin = point of the shoulder to pin bone; Heart = heart girth; Rump = width at hook bones; BMI = body mass index; BMI-W = BW/Wither [g/cm^2^]; BMI-H = BW/Hook [g/cm^2^]; BMI-P = BW/Pin [g/cm^2^]; BMI-G = BW/Heart [g/cm^2^]; BMI-WH = BW/(Wither × Hook) [g/cm^2^]; BMI-WP = BW/(Wither × Pin) [g/cm^2^]; BMI-GH= BW/(Heart × Hook) [g/cm^2^]; BMI-GP = BW/(Heart × Pin) [g/cm^2^]; BMI-WGH = BW/(Wither × Heart × Hook) [g/cm^3^].

## Data Availability

Mean data are presented in tables.

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
