# Peer review of "Effects of Dietary Inclusion of Tannin-Rich Sericea Lespedeza Hay on Relationships among Linear Body Measurements, Body Condition Score, Body Mass Indexes, and Performance of Growing Alpine Doelings and Katahdin Ewe Lambs"

_animals, 2022, doi:10.3390/ani12223183_

Round 1

Reviewer 1 Report

The study summarizes a data set with a large number of variables, and establish correlations between each and every one of the variables, however, the authors do not clearly state the aim of the project. If the objective is to find a more objective measurement to body condition score using body mass index model than the focus should be moved way from a species comparison, and data analyzed within species.  The term ‘breed’ is used in the manuscript to describe what really is a species comparison. If the aim was to determine differences between dairy and meat-type breeds that should have been done within species. Here, species becomes a confounding effect for a breed type comparison.

There is a need to more concisely summarize the data and the large number of comparisons for each variable measured. Recommendations should be made as of the BMI model best suited to replace body condition score with a more objective measurements, and how this model relates to performances measures.

Author Response

The study summarizes a data set with a large number of variables, and establish correlations between each and every one of the variables, however, the authors do not clearly state the aim of the project. If the objective is to find a more objective measurement to body condition score using body mass index model than the focus should be moved way from a species comparison, and data analyzed within species.  The term ‘breed’ is used in the manuscript to describe what really is a species comparison. If the aim was to determine differences between dairy and meat-type breeds that should have been done within species. Here, species becomes a confounding effect for a breed type comparison.

Response: We have now used ‘species’ instead of ‘breed’ in the appropriate context. The purpose of this study was to assess relationships between BCS and body mass indexes (BMI) with animal performance, and we were also interested if such relationships would be similar for or different between dairy goats and hair sheep raised for meat.

There is a need to more concisely summarize the data and the large number of comparisons for each variable measured. Recommendations should be made as of the BMI model best suited to replace body condition score with a more objective measurements, and how this model relates to performances measures.

Response: We have now simplified the brief overview of findings in the Summary and Conclusions section. We have now more clearly addressed preferred BMI and their relationships with performance measures.

Reviewer 2 Report

This is very dense and long paper. This article investigated relationships between body condition score, body mass indexes and other linear measurements as well as three different diets in Alpine doelings and Katahdin ewe lambs. This research brings new information on the inter relationships of all these measurements, which can be useful for further scientific investigations in young sheep or goats.

As general comments,

The objective of the paper is not clearly explained, which makes it difficult to read as there is no question for the reader to refer back to. It is only when reaching the summary and conclusions of the paper that there is a hypothesis stated.

So, you would need to clarify the objective of the paper. Why did you use 2 species? why did you compare these diets ? why did you use young animals ? all of these questions would need to be clarified.

Abstract: Your abstract is too long as it should only be 200 words so it should be shortened.

I would suggest simplifying the results section and only focus on your main findings to reach the take-home message of your paper.

Also, I would suggest to clearly state the objective of your paper at the start of your abstract.

Introduction: As one of your objectives is to compare diets, it may be useful to add some context around this. Do these diets have been tested before ? why are they of interest ?

Also, why did you choose these breeds and species ? are they of specific interest ? why did you take young/growing animals ? Add some context either the introduction or in the methods to help your reader understand what is happening and why.

Methods: Some clarifications are needed in this section.

The 4 measurement periods are not well stated in the methods and why did you use 4 ? did they correspond to anything ?

You are mentioning “two sets” (lines110-111). What are these two sets ? I could not find any explanation of these sets. You will need to clarify this.

Also, I think you were missing the analysis of the Changes of BCS, BMI etc. in your statistic section. Please add this analysis.

Results & Discussion: It is good to have some headings, it makes the sections a little clearer.

In your interpretation of the results and, especially when you talk about the BCS and changes in measures, you completely leave behind the fact that you used young animals. Even if your study happened over a period of 173 days, your animals were growing, and sheep and goats are not growing exactly the same way or with the same pattern. I think it is worth mentioning, especially when you talk about BCS which can be relatively low as they tend to develop muscles before developing fat. This could be even more important if you compare a breed of dairy goat and a meat breed of sheep.

There is a reference that is used repeatedly, Liu et al. 2019. Adding different studies could be a plus, even if Liu et al. 2019 is easy to compare to.

Summary and conclusions: Good to state your hypothesis, should have been in the introduction.

This is a good summary but this is not a conclusion. You need to state your take-home message, what should I remember / keep in mind after reading your paper. What does it all mean ? did you answer your research objective ? is there any further research to do on this ?

Like the abstract, simplify the result part of your summary as your reader just read the whole section and get to what it means and your take-home message. What do you want your reader to get out with ?

Specific comments

Throughout the paper: it is not ideal to use the coefficient of correlation “r” in sentence as if it is a word, I would remove them and word your sentences differently.

Throughout the paper: avoid using words like “generally” or “usually”.

Tables: Because your tables are quite large and go over multiple pages, you will need to add to each new page the top rows of your table then your reader knows what he/she is looking at.

Please explain in your table captions what “r” is.

Results and Discussion section: remove “(e.g. )” when citing literature.

Line 39: remove “(i.e. r)”

Lines 97, 99, 106 and 111: you either need to define what “wk” and “mo” means or use the whole words

Line 119: “Diets were offered at 08:00h after collecting …” Do you mean that diets were offered at 8am or pm or 8h after collecting data ? not clear.

Lines 125 -129: You mention “end of each period”, as they were not clearly explained earlier, you will need to explain what these periods are either earlier in the methods or in your Measures section.

Lines 208 and 211: some words and citations are in bold, please fix.

Author Response

This is very dense and long paper. This article investigated relationships between body condition score, body mass indexes and other linear measurements as well as three different diets in Alpine doelings and Katahdin ewe lambs. This research brings new information on the inter relationships of all these measurements, which can be useful for further scientific investigations in young sheep or goats.

Response: Thanks for your positive evaluations.

 As general comments,

The objective of the paper is not clearly explained, which makes it difficult to read as there is no question for the reader to refer back to. It is only when reaching the summary and conclusions of the paper that there is a hypothesis stated.

So, you would need to clarify the objective of the paper. Why did you use 2 species? why did you compare these diets? why did you use young animals? all of these questions would need to be clarified.

Response: We have attempted to answer these questions in the introduction of this manuscript. We have now put forward a hypothesis and objective of this study hopefully in a relatively clear manner.

Abstract: Your abstract is too long as it should only be 200 words so it should be shortened.

I would suggest simplifying the results section and only focus on your main findings to reach the take-home message of your paper. Also, I would suggest to clearly state the objective of your paper at the start of your abstract.

Response: We have added an objective of this study to the Abstract and attempted to shorten the results presented in this section.

Introduction: As one of your objectives is to compare diets, it may be useful to add some context around this. Do these diets have been tested before? why are they of interest?

Also, why did you choose these breeds and species? are they of specific interest? why did you take young/growing animals? Add some context either the introduction or in the methods to help your reader understand what is happening and why.

Response: As stated earlier, we have tried to answer these questions in the Introduction section.  We hope that these aspects of the study are addressed in an adequately and sufficiently clear manner.

Methods: Some clarifications are needed in this section.

The 4 measurement periods are not well stated in the methods and why did you use 4? did they correspond to anything?

Response: This was addressed at L96-97 (in original manuscript). Four measurements were performed to obtain data in different times during their growth.  It is possible that such relationships could vary with growth and associated changes in body composition.  The text has been revised in this regard.

You are mentioning “two sets” (lines110-111). What are these two sets? I could not find any explanation of these sets. You will need to clarify this.

Response: This is addressed at the end of the first paragraph of the Materials and Methods section.  The two sets were started at different times in regard to the number of animals for which data could be collected at one period of time.

Also, I think you were missing the analysis of the Changes of BCS, BMI etc. in your statistic section. Please add this analysis.

 Response: We mentioned “Pearson correlation coefficients (r) were used to evaluate relationships between variables [30]” which also includes the correlations for the changes. We have revised the text in this regard.

Results & Discussion: It is good to have some headings, it makes the sections a little clearer.

Response: This section has a few headings to make the section clearer and easier to follow.

In your interpretation of the results and, especially when you talk about the BCS and changes in measures, you completely leave behind the fact that you used young animals. Even if your study happened over a period of 173 days, your animals were growing, and sheep and goats are not growing exactly the same way or with the same pattern. I think it is worth mentioning, especially when you talk about BCS which can be relatively low as they tend to develop muscles before developing fat. This could be even more important if you compare a breed of dairy goat and a meat breed of sheep.

Response: These are good suggestions. We have introduced this view in the Introduction and Discussion sections.

There is a reference that is used repeatedly, Liu et al. 2019. Adding different studies could be a plus, even if Liu et al. 2019 is easy to compare to.

Response: We agree.  We have added many other relevant references.  But, the Liu et al. (2019) study was very important in terms of being the first comprehensive evaluation of these linear measures and the large number of BMI.

 Summary and conclusions: Good to state your hypothesis, should have been in the introduction.

Response: We have now stated a hypothesis in the Introduction section.

This is a good summary but this is not a conclusion. You need to state your take-home message, what should I remember / keep in mind after reading your paper. What does it all mean? did you answer your research objective? is there any further research to do on this?

Like the abstract, simplify the result part of your summary as your reader just read the whole section and get to what it means and your take-home message. What do you want your reader to get out with?

 Response: We have revised the Summary and Conclusions section.

Specific comments

Throughout the paper: it is not ideal to use the coefficient of correlation “r” in sentence as if it is a word, I would remove them and word your sentences differently.

Response: We have revised the manuscript in regard to these comments.

Throughout the paper: avoid using words like “generally” or “usually”.

Response: We used “generally” two times and “usually” three times to indicate most incidences (but not all cases) to summarize them.  Because these words are used to indicate that the great majority of relationships existed as mentioned, but not all or 1000% of them fit closely with the statement, use of these words is considered appropriate.  And, to delete them would result in a less accurate description of the findings than is desired. 

Tables: Because your tables are quite large and go over multiple pages, you will need to add to each new page the top rows of your table then your reader knows what he/she is looking at.

Response:  These comments are understood.  Currently the part of the table that will appear on a new page cannot be accurately predicted.  This will depend on final formatting of the journal.    Nonetheless, from our previous experiences with this journal, the copy editor will adequately address this issue.

Please explain in your table captions what “r” is.

Response:  This was stated in the table caption as an association (r). Now, it has been defined as a Pearson correlation coefficient(r).

Results and Discussion section: remove “(e.g. )” when citing literature.

Response:  This has been removed, although the intent was to signify that other citations are available for listing.

Line 39: remove “(i.e. r)”

Response: We have deleted ‘i.e.,’ as r has been used in the Abstract to indicate correlation coefficients.

Lines 97, 99, 106 and 111: you either need to define what “wk” and “mo” means or use the whole words.

Response: Wk has been defined when used for the first time. Month has been  substituted for mo as they were used two times.  However, for many journals ‘wk’ and ‘mo’ are standard unit abbreviations not requiring definition.

Line 119: “Diets were offered at 08:00h after collecting …” Do you mean that diets were offered at 8am or pm or 8h after collecting data? not clear.

Response:  It is a standard practice to present times in this manner (i.e., over 24-h periods).  This indicates that animals were offered the diet at 8 am.

Lines 125 -129: You mention “end of each period”, as they were not clearly explained earlier, you will need to explain what these periods are either earlier in the methods or in your Measures section.

Response: It was mentioned that “It consisted of four measurement periods, the first three 6 wk in length and the last 47 days.” (L 96-97). We have now also modified the text here for improved clarity.

Lines 208 and 211: some words and citations are in bold, please fix.

Response:  There were typographical errors, and they have been corrected now.

Round 2

Reviewer 1 Report

The authors address the issue that the study is not truly a breed comparison, actually a species comparison by replacing every mention of ‘breed’ with ‘species’.  However, it would have been useful to also change the use of Alpine (ALP) and Katahdin (KAT) to ‘sheep’ and ‘goat’, to reflect the species rather than breed comparison. 

While I feel that the species comparison is somewhat confounded by comparing a dairy-type goat breed with a meat-type sheep breed, a fact is now acknowledged by the authors in the discussion (lines 189-192 describes differences between diary and meat type sheep breed in muscle and fat deposition), and the initial comparison has to remain species. The authors should extend their effort to clearly describe species differences as such, while acknowledging the confounding effect of breed type inherent to the study. A good initial step the reference to sheep breeds differing in their correlation between BCS and BMI (lines 234-238), and could be extended to other parts of the discussion.

The following minor edits are needed in regard to mention of breed:

Line 46: change ‘breed’ to ‘species’

Line 180: delete ‘breed and’

Line 393: ‘breed x diet’ should be ‘species x diet’

Author Response

The authors address the issue that the study is not truly a breed comparison, actually a species comparison by replacing every mention of ‘breed’ with ‘species’.  However, it would have been useful to also change the use of Alpine (ALP) and Katahdin (KAT) to ‘sheep’ and ‘goat’, to reflect the species rather than breed comparison. 

Response: We have initially mentioned that Alpine (ALP) goats and Katahdin (KAT) sheep were used. In this way, readers will easily understand that ALP and KAT denote Alpine goats and KAT sheep, respectively and simultaneously they will understand the breeds of  the animals.

While I feel that the species comparison is somewhat confounded by comparing a dairy-type goat breed with a meat-type sheep breed, a fact is now acknowledged by the authors in the discussion (lines 189-192 describes differences between diary and meat type sheep breed in muscle and fat deposition), and the initial comparison has to remain species. The authors should extend their effort to clearly describe species differences as such, while acknowledging the confounding effect of breed type inherent to the study. A good initial step the reference to sheep breeds differing in their correlation between BCS and BMI (lines 234-238), and could be extended to other parts of the discussion.

Response: Thanks for the good suggestions. We have mentioned about the confounding of animal type and species in this part and conclusion section. We have also referred about species difference for the muscle and noncarcass fat deposition. However, the literature comparing directly between sheep and goat species for the relations of BCS and BMI is lacking.

The following minor edits are needed in regard to mention of breed:

Line 46: change ‘breed’ to ‘species’

Response: Thanks. We have corrected it.

Line 180: delete ‘breed and’

Response: Thanks. We have corrected it.

Line 393: ‘breed x diet’ should be ‘species x diet’

Response: Thanks. We have corrected it.

Reviewer 2 Report

The authors have clarified the manuscript in regard to my recommendations. The aim and the summary of the study are clearer.

Author Response

The authors have clarified the manuscript in regard to my recommendations. The aim and the summary of the study are clearer.

Response: Thanks for the useful suggestions that helped to improve the quality of this manuscript.